

# Locating and quantifying $CH_4$ sources within a wastewater treatment plant based on mobile measurements

Junyue Yang[1], Zhengning Xu[1], Zheng Xia[4,5], Xiangyu Pei[1], Yunye Yang[1], Botian Qiu[2,3], Shuang Zhao[2,3], Yuzhong Zhang[2,3*], Zhibin Wang[1,6*]

[1]Zhejiang Provincial Key Laboratory of Organic Pollution Process and Control, MOE Key Laboratory of Environment Remediation and Ecological Health, College of Environmental and Resource Sciences, Zhejiang University, Hangzhou 310058, China

[2]Key Laboratory of Coastal Environment and Resources of Zhejiang Province, School of Engineering, Westlake University, Hangzhou 310030, China

[3]Institute of Advanced Technology, Westlake Institute for Advanced Study, Hangzhou 310024, China

[4]Ecological and Environmental Monitoring Center of Zhejiang Province, Hangzhou 310012, China

[5]Zhejiang Key Laboratory of modern Ecological and Environmental Monitoring, Hangzhou 310012, China

[6]ZJU-Hangzhou Global Scientific and Technological Innovation Center, Zhejiang University, Hangzhou 311200, China

*Correspondence to*: Zhibin Wang (wangzhibin@zju.edu.cn) and Yuzhong Zhang (zhangyuzhong@westlake.edu.cn)

**Abstract.** Wastewater treatment plants (WWTPs) are substantial contributors to greenhouse gas (GHG) emission because of the high production of methane ($CH_4$) and nitrous oxide ($N_2O$). A typical WWTP complex contains multiple functional areas that are potential sources for GHG emissions. Accurately quantifying GHG emissions from



these sources is challenging due to the inaccuracy of emission data, the ambiguity of
emission sources, and the absence of monitoring standards. Locating and quantifying
WWTPs emission sources in combination with measurement-based GHG emission
quantification methods are crucial for evaluating and improving traditional emission
inventories. In this study, $CH_4$ mobile measurements were conducted within a WWTP
complex in the summer and winter of 2023. We utilized a multi-source Gaussian plume
model combined with the genetic algorithm inversion framework, designed to locate
major sources within the plant and quantify the corresponding $CH_4$ emission fluxes. We
identified 12 main point sources in the plant and estimated plant-scale $CH_4$ emission
fluxes of $603.33 \pm 152.66$ t a$^{-1}$ for the summer and $418.95 \pm 187.59$ t a$^{-1}$ for the winter.
The predominant sources of $CH_4$ emissions were the screen and primary clarifier,
contributing 55 % and 67 % to the total emissions in summer and winter, respectively.
The comparison against traditional emission inventories revealed that the $CH_4$ emission
fluxes in the summer were 2.8 times greater than the inventory estimates, and in the
winter, emissions were twice the inventory values. Our flux inversion method achieved
a good agreement between simulations and observations (correlation > 0.6 and a root
mean square error (RMSE) < 0.7 mg m$^{-3}$). This study demonstrated that mobile
measurements, combined with the multi-source Gaussian plume inversion framework,
are a powerful tool to locate and quantify GHG sources in a complex site, with the
potential for further refinement to accommodate different types of factories and gas
species.

## 1 Introduction

Greenhouse gas (GHG) emissions exacerbate the greenhouse effect, causing adverse
impacts on human health, ecosystems, and the environment (IPCC, 2023). Methane
($CH_4$) is the second-largest contributor to climate change, with the global warming
potential 27.9 times that of carbon dioxide ($CO_2$). Reducing $CH_4$ emissions is essential



for mitigating climate change and progressively achieving the global target of limiting warming to 1.5 °C. The latest observational study from the WMO Global Atmospheric Watch network indicated that the global annual average concentration of $CH_4$ in 2022 was 1923 ± 2 ppb, representing a 264 % increase from pre-industrial levels (WMO, 2023). The International Energy Agency (IEA) 's 2024 Global Methane Tracker report suggests that global $CH_4$ emissions reached 580 Mt in 2023, with anthropogenic $CH_4$ emissions accounting for 60 %. The complexity of $CH_4$ emission processes, lack of monitoring systems, and limitations of emission estimation models present challenges in accurately estimating anthropogenic $CH_4$ emissions.

The quantification of $CH_4$ emission fluxes is typically achieved through a bottom-up inventory method. However, due to the difficulties in obtaining actual emission factors activity data, and specific information on different emission sources, there is considerable uncertainty in the assessing of the emission inventory method (Lin et al., 2021). In contrast, a top-down method that estimates $CH_4$ emissions by monitoring atmospheric concentration has been increasingly applied in recent years (Sun et al., 2019; Cusworth et al., 2024; Han et al., 2024; Maazallahi et al., 2023; Riddick et al., 2017). The monitoring technology mainly includes satellite (Zhang et al., 2021; Liang et al., 2023; Jacob et al., 2022) and airborne (Allen et al., 2019; Abeywickrama et al., 2023; Cui et al., 2017) remote sensing, as well as ground-based monitoring such as vehicle-based mobile monitoring (Albertson et al., 2016; Al-Shalan et al., 2022; Caulton et al., 2018), station monitoring (Dietrich et al., 2021; Hase et al., 2015; Heerah et al.,2021) and tower monitoring (Richardson et al., 2017; Balashov et al., 2020). Numerous studies use satellite remote sensing, unmanned aerial vehicle (UAV) monitoring, and vehicle-based mobile monitoring techniques to measure $CH_4$ emissions (Sun et al.,2023). However, satellite spatiotemporal resolution is limited and UAVs have short endurance, making vehicle-based mobile monitoring a better choice for measuring emissions at wastewater treatment plants (WWTPs). Vehicle-based mobile monitoring can perform continuous real-time monitoring and precise identification of



emission sources, and hence have been applied to urban (von Fischer et al., 2017; Defratyka et al., 2021) and plant-scale (Zhao et al., 2021; Jin et al.,2010) monitoring of GHG concentrations and emission fluxes. Vogel et al. (2024) investigated $CH_4$ leaks in 12 cities across 8 countries, using high-precision fast-response GHG analyzers combined with the mobile survey methodology (von Fischer et al., 2017). Chen et al. (2020) utilized the multiple-Gaussian-plume model and a forward modeling approach for mobile measurements of $CH_4$ emissions during the Munich Oktoberfest. Shi et al. (2023) proposed a $CO_2/CH_4$ emission quantification model (EMISSION-PARTITION) and conducted mobile measurements with vehicle-based monitoring system at chemical, coal washing, and waste incineration plants in two cities and one industrial park in China, assuming different numbers of emission sources for quantitative assessment. Wang et al. (2022a; 2022b) employed the Environmental Protection Agency's Other Test Method 33A (OTM 33A) for monitoring downwind of fueling stations to estimate the $CH_4$ emission fluxes of nine compressed natural gas (CNG) stations and five liquefied natural gas (LNG) stations in Eastern China. Emission flux inversion methods also include isotope tracer method (Jackson et al., 2014; Zimnoch et al., 2018), cross-sectional flux method (Luther et al., 2019; Makarova et al., 2021), and atmospheric diffusion model inversion method (Kumar et al., 2021; Yacovitch et al., 2015). Atmospheric transport models with varied degrees of complexity, including Gaussian diffusion models (Stadler et al., 2021), Lagrangian models (Mckain et al., 2015), and Eulerian models (Bergamaschi et al., 2018), are used in the inversion to relate GHG concentrations with emissions. Optimization methods, such as Bayesian optimization (Karion et al., 2019) and linear regression models (Kumar et al., 2021), are applied to achieve accurate inversion results. Furthermore, some studies incorporate carbon isotope observations to better attribute the contribution of different $CH_4$ emission sources (Maazallahi et al., 2020).

As a significant source of GHG emissions, WWTPs generate substantial amounts of $CH_4$, $N_2O$, and $CO_2$ during the collection, treatment, and discharge of sewage and



sludge, contributing 3 % of the global total GHG emissions (Bai et al., 2022). The estimation of $CH_4$ emission fluxes from WWTPs has increasingly attracted widespread attention. Li et al. (2024) developed a plant-level and technology-based $CH_4$ emission inventory for municipal WWTPs in China, estimating the $CH_4$ emissions for 2020 to be 150.6 Gg. Wang et al. (2022) systematically considered process technological differences in wastewater treatment, constructing a high-resolution greenhouse gas emission inventory for Chinese WWTPs from 2006 to 2019. Delre et al. (2017) measured the $CH_4$ and $N_2O$ concentrations downwind of five WWTPs in Scandinavia using tracer gas dispersion, which obtained a range of $CH_4$ emission fluxes from 1.1 ± 0.1 to 18.1 ± 6.3 kg h$^{-1}$. Moore et al. (2023) employed an integrated Gaussian plume model with a Bayesian framework for mobile measurements of $CH_4$ emissions from 63 WWTPs in the United States, pointing to a significant underestimation in the $CH_4$ emission inventories.

We present a mobile measurement investigation of a WWTP in Hangzhou 2023. To analyze the mobile data, we construct a multi-source Gaussian plume model combined with the genetic algorithm inversion framework, which assists us to locate and quantify $CH_4$ emission sources, based on the concentration distribution measured within the WWTP. Additionally, we compare $CH_4$ emission fluxes from the measurements with the bottom-up estimates of emission inventories. A sensitivity analysis is performed to elucidate the discrepancies arising from variations in emission source locations. Our results provide insight into formulating and evaluating emission reduction measures for WWTPs.

## 2 Instruments and methods

### 2.1 Site selection

The monitoring site was chosen at a WWTP in Hangzhou, a megacity in East China.



This WWTP is a large-scale plant located in Hangzhou, processing up to 1.5 million
tons of domestic wastewater daily. The plant roads were flat and wide, suitable for
vehicle-mounted CRDS (Cavity Ring-Down Spectroscopy) to conduct monitoring
along the internal roads of the plant to monitor various functional areas within the plant.
WWTPs processes typically encompass mechanical treatment, biological treatment,
sedimentation, advanced treatment, disinfection, and sludge treatment. As illustrated in
Fig. 1, we divide the WWTP into 14 functional areas according to treatment processes.
For instance, areas associated with primary treatment were labeled as coarse screens
and primary sedimentation tanks, while those linked to secondary treatment were
indicated as aeration tanks and secondary sedimentation tanks. Mobile measurements
were conducted by driving around the outer periphery and internal functional areas of
the wastewater treatment plant, with each monitoring experiment involving circling the
functional areas 1-2 times. 10 days of experiments were carried out from June to
December 2023. This yielded 8 valid sets of monitoring data, including 3 days of
summer data and 5 days of winter data.

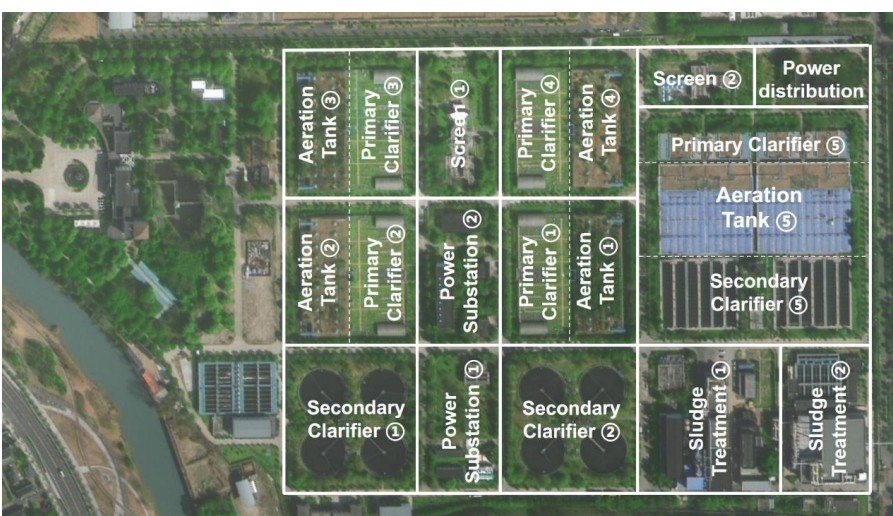

**Figure 1.** Distribution of functional areas of the WWTP. Map data are from ESRI.



**2.2 Instrumentation**

The monitoring instruments consisted of a vehicle-mounted CRDS monitoring system and a portable meteorological station. The vehicle-mounted CRDS system was anchored by the CRDS analyzer (Picarro G2201-i), accompanied by GPS and meteorological instruments. The CRDS analyzer measures $^{12}CO_2$, $^{13}CO_2$, $^{12}CH_4$, $^{13}CH_4$ and $H_2O$, the volume fraction of $CH_4$ is measured with an accuracy of 5 ppb $\pm$ 0.05 % (Picarro 2010). CRDS measurements have the advantages of strong interference resistance, high sensitivity and accuracy, making them widely employed in research focused on monitoring GHG emissions (Rella et al., 2015; Lopez et al., 2017). In this study, the CRDS analyzer was securely placed inside the monitoring vehicle, with the sampling probe mounted on the roof to mitigate the effects of vehicular emissions. The system was powered by a battery, drawing in ambient air through a pump, and displaying real-time monitoring data on a screen. The mobile meteorological instrument was placed on the roof of the vehicle to gather meteorological data. In addition, the GPS unit was integrated to record the location of sampling points during the measurement period.

Two portable meteorological stations (SWS-500) were positioned adjacent to the main entrance and atop the filter tank at the WWTP. Capable of measuring key meteorological parameters such as wind speed, direction, temperature, humidity, and atmospheric pressure, this station provided essential climatic data for the monitoring experiments. Mobile measurements were performed by the monitoring vehicle along the entire roads of the WWTP, as well as the internal roads, to pinpoint the locations of emission sources, scrutinize variations in emission concentrations. The concentration data was subsequently integrated with an inversion model to estimate the $CH_4$ emission fluxes.



### 2.3 Inventory accounting method

We used the methods suggested by the IPCC Guidelines for National Greenhouse Gas Inventories (2006) to calculate the amounts of $CH_4$ emissions from wastewater. The formula for calculating the amounts of $CH_4$ emissions from wastewater is described as:

$$E_{CH_4} = (TOW - S \cdot a) \cdot EF_{CH_4} - R_{CH_4} \tag{1}$$

Where $E_{CH_4}$ denotes the direct $CH_4$ emissions from the wastewater treatment plant, $tCH_4$ a$^{-1}$. $TOW$ is defined as the total organic pollutant load in the influent wastewater, $tCOD$ a$^{-1}$. $S$ refers to the annual production total of dry sludge, t a$^{-1}$. The parameter $a$ signifies the organic matter content in the dry sludge, $tCOD$ t$^{-1}$. $EF_{CH_4}$ is the $CH_4$ emission factor, $tCH_4$ $tCOD^{-1}$. $R_{CH_4}$ quantifies the annual recovery of $CH_4$ from anaerobic treatment processes, t a$^{-1}$.

Operational data of the WWTP examined in this study is derived from the Urban Drainage Statistical Yearbook, an annual publication of urban water supply and drainage systems in China. This data set includes details such as the water treatment volume, sludge production, and the concentrations of six pollutants ($COD_{Cr}$, BOD, SS, $NH_3$-N, TN, and TP) in both influent and effluent. The Total Organic Waste (TOW) is deduced from the yearbook's foundational data, while the annual sludge production (S) is extracted directly from it. The organic matter content in dry sludge is estimated at an empirical 40 %, assuming a sludge moisture content of 75 %, leading to a value of 0.1 (Guo et al., 2019). $EF_{CH_4}$ is selected based on the recommended value for Zhejiang province, 0.0046 (Cai et al., 2015). Given the infrequency of anaerobic treatment in wastewater, $R_{CH_4}$ is set to 0.

### 2.4 Inversion method

We developed an inversion framework for $CH_4$ emission fluxes designed for plant-



level applications. The framework used $CH_4$ concentration measurements, specific
locations of emission sources, and initial emission estimates, alongside wind speed and
direction data, as inputs to the multi-point and line source Gaussian diffusion models.
The preliminary localization of the emission sources was chiefly contingent upon the
concentration distribution along the roads within the internal functional areas.
Meanwhile, the initial emission estimates for each source were determined by
integrating the concentration data from these areas with an improved empirical equation
(Weller et al., 2018). These inputs were fed into a multi-source Gaussian plume model
that simulates the concentration patterns of $CH_4$ given multiple point and line sources.
We then used a genetic algorithm to iteratively optimize source emission fluxes and
their locations. The inversion framework simulation dictated the placement of 12 main
point sources throughout the WWTP, specifically within Aeration Tank ①②③④⑤,
Primary Clarifier ③④⑤, Screen ①, Secondary Clarifier ①②, and the Sludge
Treatment ② (Fig. 1). The inclusion of a Gaussian line source model was determined
based on the actual emission conditions. Within this study, a uniform line source was
established, with the assumed location along the road between the Screen ① and the
Primary Clarifier ①. This assumption was grounded in the $CH_4$ concentration
distribution observed within this road segment and was substantiated through model
validation, confirming the existence of a line source emission pattern. The remaining
emission flux inversion processes followed the same procedure as the point source
simulation. Adjustments to the source locations within the model narrow the gap
between simulated and measured concentrations, thus enhancing the accuracy of
inversion. This section delineates each model incorporated into the inversion
framework.



### 2.4.1 Multiple-point-source Gaussian plume model

We developed a multiple-point-source Gaussian plume model to relate $CH_4$ concentration enhancement to $CH_4$ emissions. This method approximates atmospheric dispersion of $CH_4$ from an individual source as a Gaussian plume under uniform and stable wind conditions (Nassar et al., 2017), which is usually good for describing average atmospheric transport tens to hundreds of meters downwind the source, making the Gaussian plume model a useful tool to study emissions from industrial and traffic sources.

The mass concentration enhancement ($C$, mg m$^{-3}$) is computed as superposition of Gaussian plumes from multiple point sources.

$$C(x, y, z) = \sum_{i=1}^{n} \frac{Q_i}{2\pi \bar{u} \sigma_{i,y} \sigma_{i,z}} exp\left(-\frac{(y-y_i)^2}{2\sigma_{i,y}^2}\right) \left\{ exp\left[\frac{-(z-z_i)^2}{2\sigma_{i,z}^2}\right] + exp\left[\frac{-(z+z_i)^2}{2\sigma_{i,z}^2}\right] \right\} \quad (2)$$

The variables $x$, $y$, and $z$ denote the downwind, crosswind distances, and the height above the ground from the source, m. $Q_i$ signifies the emission rate from the $i_{th}$ point source, mg/s, for $i$ = 1, 2, 3……N, where $N$ represents the total count of point sources. The average wind speed is indicated by $\bar{u}$, m s$^{-1}$. The $x_i$, $y_i$ and $z_i$ are represented as the spatial position of the $i_{th}$ point source, m. $\sigma_{i,y}$ and $\sigma_{i,z}$ are the horizontal and vertical dispersion parameters of the $i_{th}$ point source, respectively, which are given by the formula below:

$$\sigma_{i,y} = \gamma_1 \cdot (x - x_i)^{\alpha_1}, \ when \ x > x_i \quad (3)$$

$$\sigma_{i,z} = \gamma_2 \cdot (x - x_i)^{\alpha_2}, \ when \ x > x_i \quad (4)$$

The power functions, known as the Pasquill's curves, associates with the downwind distance x and the prevailing atmospheric stability (Briggs et al., 1973). Atmospheric stability is determined based on the Pasquill stability classes recommended in the Technical Principles and Methods for Formulating Local Air Pollution Emission Standards (GB3840-83).



### 2.4.2 General Finite Line Source Model


Our analysis of measurement at WWTPs indicates that multiple-point-source
Gaussian plume model is insufficient to capture the observed CH$_4$ concentrations. The
entire road between the Screen ① and the Primary Clarifier ① shows high
distribution of CH$_4$ concentrations. To match the observations, we further consider a
line source based on observed concentration distribution. The line source model is used
to confirm that the road concentration distribution is consistent with line source
emissions (Fig. S1). The contribution of a line source to CH$_4$ concentration is given by
the General Finite Line Source Model (GFLSM) (Luhar et al., 1989; Venkatram et al.,
2006), which represents the line source as an ensemble of point sources:

$$C = \frac{Q}{2\pi\bar{u}\sigma_y\sigma_z}\left\{exp\left[\frac{-(z-H)^2}{2\sigma_z{}^2}\right] + exp\left[\frac{-(z+H)^2}{2\sigma_z{}^2}\right]\right\}$$

$$\cdot\left[erf\left(\frac{\sin\theta\left(\frac{L}{2}-y\right)-x\cos\theta}{\sqrt{2}\sigma_y}\right) + erf\left(\frac{\sin\theta\left(\frac{L}{2}+y\right)+x\cos\theta}{\sqrt{2}\sigma_y}\right)\right] \tag{5}$$


$x$, $y$, and $z$ correspond to the downwind, crosswind distances, and the altitude above
ground level from the source, m. $Q_i$ is the emission fluxes of the unit source, mg s$^{-1}$. $\bar{u}$
is the average wind speed, m s$^{-1}$. $H_i$ is the effective emission height of the line source,
with the length of the line source represented by $L$, m. The angle between the line
source and the wind direction is given by $\theta$. The horizontal and vertical dispersion
parameters are characterized by $\sigma_y$ and $\sigma_z$, respectively.

### 2.4.2 Genetic algorithm


Genetic algorithms, which mimic the evolutionary process of biological systems,
serve as optimization search algorithms. The algorithms encode practical problems into
binary genetic coding. Through the simulation of natural selection, crossover, and
mutation processes, these algorithms are in a constant state of evolution and iteration,



all in the pursuit of the optimal solution (Katoch et al., 2021). We deployed genetic
algorithms to enhance the source emission flux outcomes modeled by the Gaussian
plume model.
The process of inverting multi-source $CH_4$ emission fluxes utilizing genetic
algorithms involves a series of steps. Initially, the emission flux of each source is treated
as a gene, with binary-encoded gene sequences randomly assigned to a set number of
individuals within the predefined range of a priori emission fluxes. Subsequently, the
formulation of a fitness function is based on the defined optimization goals and
constraints. This function serves as a critical tool for assessing the relative merits of
each individual within the population. In this study, the objective of the optimization is
centered on minimizing the aggregate absolute discrepancy between the values
predicted by the model and those obtained from measurements. Ultimately, the
population is subjected to the processes of selection, crossover, and mutation.
Individuals with elevated fitness values, as determined by the fitness function, are
chosen for the generation of new individuals. Through an iterative process, the optimal
solution is refined, representing the emission fluxes for each source. Genetic algorithms
are distinguished by the parallel computation capabilities, the propensity for identifying
global optima, and the commendable stability and reliability (Harada et al., 2020).

**3 Results and discussion**
**3.1 Concentration mapping**
The closed-path mobile measurements were conducted by vehicle-mounted CRDS
monitoring system along the external roads encircling the WWTP, with further
monitoring conducted along the internal roads. This strategy depicts the distribution of
$CH_4$ concentrations within an WWTP, allowing for identification of specific $CH_4$
emission sources. Based on 8 days of $CH_4$ monitoring experimental data, the $CH_4$



concentration range on the overall roads was determined to be 1.98-17.13 ppm. The $CH_4$ concentration distribution indicated higher levels downwind, with the highest concentrations consistently recorded at the Screen ① throughout mobile experiments.

Due to the similarity of concentration measurement methods, we chose 29[th] June and 13[th] December as a typical example for measuring the spatial distribution of $CH_4$ and evaluating the seasonal variability of WWTP. Figure 2 illustrates measured $CH_4$ concentration enhancement distributions on 29[th] June (summer) and 13[th] December (winter) 2023 (other days are shown in Figures S2-S7). The $CH_4$ concentration enhancements depicted within the figures were calculated by subtracting the background concentrations from the measured values, with the background determined as the mean of the bottom 10 % of the concentration data. Specifically, the background concentrations register at 1.98 ppm on 29[th] June and at a slightly elevated 2.11 ppm on 13[th] December. Moreover, increased concentrations are detected in the regions surrounding the Screen ①, Primary Clarifier ④, and Aeration Tank ③ during these two days. The complete concentration maps , which include the internal roads, reveal that the experiment on 29[th] June exhibits heightened concentrations at the Screen ①, Secondary Clarifier ②, and Primary Clarifier ②④. The Screen ① exhibits the highest $CH_4$ concentration, with an enhancement of 14.83 ppm. On 13[th] December, the concentration enhancements are noted in proximity to the Secondary Clarifier ① and Primary Clarifier ② , with the Primary Clarifier ② showing the highest $CH_4$ concentration at 4.79 ppm.

$CH_4$ concentrations in summer surpass those observed in winter, consistent with a previous study on WWTPs (Masuda et al., 2015). The screen, primary clarifier and aeration tank are identified as sources with notably higher concentrations. Analysis of concentration distributions reveals that Screen ① shows a peak concentration reaching 14.83 ppm, which is 7.5 times the background concentration. The four primary



clarifiers record high concentrations between 4.79 and 10.88 ppm. The high value
measured by aeration tanks is mainly detected in Aeration Tank ③ at 4.60ppm. The
screen in this study includes coarse and fine screens and a grit chamber, constituting
preliminary wastewater treatment to capture larger suspended solids and particulates.
The anaerobic environment of the sewer network promotes the production of $CH_4$ from
organic compounds in municipal wastewater. As this wastewater enters the WWTP, the
influent contains dissolved $CH_4$ that originated in the sewer network. During primary
treatment, wastewater is elevated through riser mains, facilitating the release of $CH_4$
into the atmosphere (Guisasola et al., 2008; Bao et al., 2016). Flow velocity, hydraulic
design and detention times in these facilities may affect $CH_4$ production and release
(Alshboul et al., 2016; Yin et al., 2024). The primary clarifier physically removes
suspended solids from wastewater through sedimentation, while organic matter
undergoes anaerobic microbial degradation to the substantial production of $CH_4$
(Masuda et al., 2017). In the aeration tank, operated under anaerobic and anoxic
conditions, complex organic compounds are converted to $CH_4$ by facultative and
anaerobic bacteria through biological processes (Yoshida et al.,2014). In contrast,
Kupper et al. (2018) identified sludge storage tanks as the primary source of $CH_4$
emissions in Swiss WWTPs, accounting for 70 % or more of the total emissions. Stadler
et al. (2022) monitored $CH_4$ concentrations inside and around wastewater treatment
facilities ranging from 2.04-32.78 ppm, with elevated $CH_4$ levels predominantly
measured near sludge treatment tank, the digesters and secondary clarifiers.



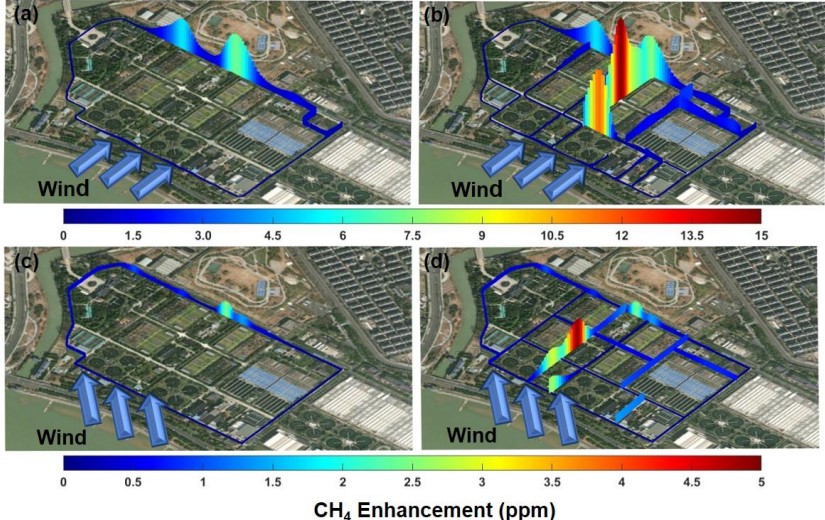


**Figure 2.** CH$_4$ concentration maps in the WWTP. The concentration maps for the external roads for 29$^{th}$ June (a) and 13$^{th}$ December (c). The corresponding complete concentration maps that include the internal roads for 29$^{th}$ June (b) and 13$^{th}$ December (d). Map data are from ESRI.


**3.2 Emission quantification**

The mobile measured CH$_4$ concentrations were employed in combination with the
inversion framework to achieve the quantification of CH$_4$ emissions and localization of
the emission sources within the WWTP. Figures 3 and 4 show the locations of identified
point sources and the comparison between monitored and simulated concentrations for
the point source locations at the WWTP on the dates of 29$^{th}$ June and 13$^{th}$ December.
The experiment conducted on 29$^{th}$ June finds the Screen ① to be the most significant
contributor to CH$_4$ point source emissions at 160.19 t a$^{-1}$, and the Secondary Clarifier
② as the least significant at 10.78 t a$^{-1}$. The correlation coefficient R$^2$ for the monitored
and simulated concentrations is 0.63, with an RMSE of 0.70 mg m$^{-3}$. On 13$^{th}$ December,





the Aeration Tank ⑤ is the largest point source of $CH_4$ emissions at 34.48 t $a^{-1}$, and
the Primary Clarifier ⑤ is the smallest at 4.82 t $a^{-1}$, with a correlation coefficient $R^2$
of 0.70 and an RMSE of 0.28 mg $m^{-3}$. The enhanced correlation between winter
monitoring and simulation data, as well as the improved fit of the monitoring and
simulation value curves, is attributed to the shorter monitoring cycle and more stable
meteorological conditions.

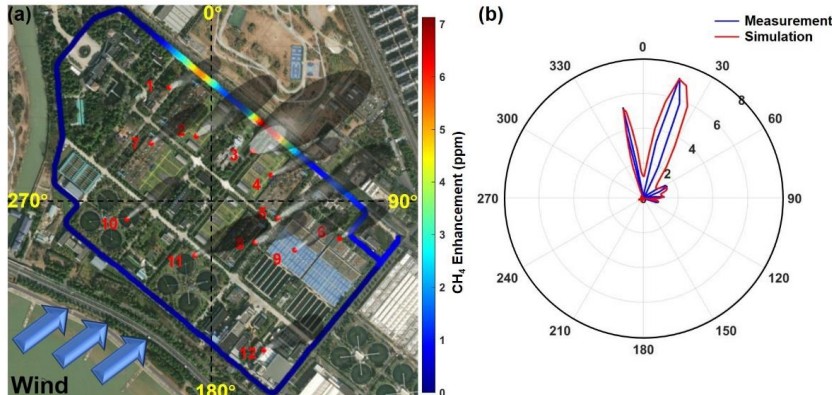


**Figure 3.** The emission distribution for the source locations (a) and the comparison
between monitored and simulated $CH_4$ concentrations (b) at the WWTP on 29th June.
Map data are from ESRI.

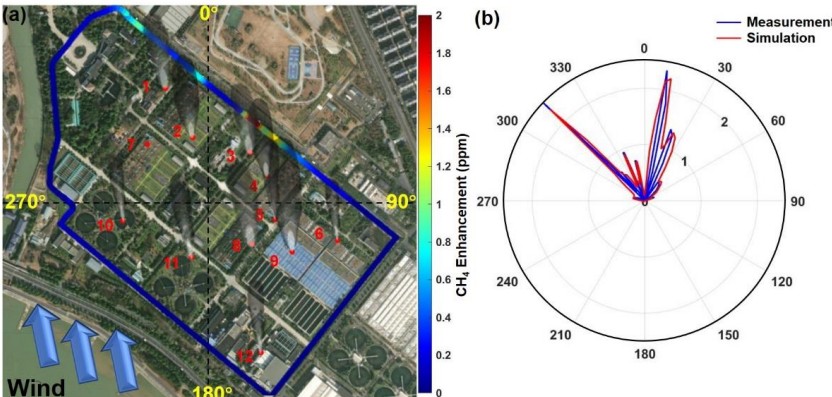


**Figure 4.** The emission distribution for the source locations (a) and the comparison



between monitored and simulated $CH_4$ concentrations (b) at the WWTP on 13[th]
December. Map data are from ESRI.

Table 1 displays the $CH_4$ emission fluxes, meteorological data, and the coefficients
of the power function expressions for diffusion parameters from the 8-day monitoring
experiment. The emission flux values of $CH_4$ emission sources (12 point sources and 1
line source) for all experimental days are detailed in Tables S1 and S2. It is observed
that the summer average $CH_4$ emission flux $(603.33 \pm 152.66$ t a$^{-1})$ surpasses the winter
average $CH_4$ emission flux $(418.95 \pm 187.59$ t a$^{-1})$. This seasonal disparity in emissions
is primarily attributed to the aeration tank, followed by the screen and primary clarifier.
The activated sludge in the aeration tank contains a higher population of methanogens,
whose $CH_4$ production capability intensifies with rising temperatures (Vítěz et al.,
2020). Notably, the seasonal variance in the aeration tank is predominantly driven by
the performance of the Aeration Tank ④. However, the substantial variation in the
emissions from the three summer experiments of the Aeration Tank ④ suggests a
degree of emission instability. Conversely, the uniformity in the low emissions from the
five winter experiments might be associated with the meteorological conditions and the
actual operational status of the plant on those days.
Analysis of emission source data from Tables S1 and S2 reveals that the screen and
primary clarifier are the predominant emission sources at the WWTP. Specifically, these
sources emit 329 t a$^{-1}$ in the summer and 280 t a$^{-1}$ in the winter, accounting for 55 %
and 67 % of the total emissions. The study hypothesizes that emissions are boosted by
pipeline leaks near the emission sources in the screen and primary clarifier, leading to
more $CH_4$ release. Previous research has similarly examined major emission sources at
WWTPs. Yin et al. (2024) conducted offline monitoring of WWTPs in Beijing and
Guiyang, identifying the primary treatment zone as the primary source of $CH_4$,
accounting for 60.1 % and 35.8 % of the respective total emissions. Masuda et al. (2017)



analyzed CH$_4$ emissions from different processes at three WWTPs in Japan, concluding
that primary clarifiers are one of the major sources of CH$_4$ emissions. He et al. (2023)
compiled CH$_4$ emission proportions for different processes in WWTPs based on
reported data, finding percentages of 7 %-12 % for grit chamber, 8.2 %-68.1 % for
primary clarifier, and 18.3 %-86.4 % for aeration tank.

**Table 1.** CH$_4$ emission fluxes, meteorological data and diffusion parameter power
function expression coefficients from the 8-day monitoring experiment.

| Date | Q (t a$^{-1}$) | W$_s$ (m s$^{-1}$) | W$_d$ (°) | $\gamma_1$ | $\alpha_1$ | $\gamma_2$ | $\alpha_2$ |
|---|---|---|---|---|---|---|---|
| 0601 | 542.50 ± 179.03 | 2.3 | 248.5 | 0.28 | 0.91 | 0.13 | 0.94 |
| 0629 | 657.18 ± 308.88 | 1.9 | 238.3 | 0.28 | 0.91 | 0.13 | 0.94 |
| 0711 | 610.31 ± 286.85 | 0.9 | 225.8 | 0.28 | 0.91 | 0.13 | 0.94 |
| 1213 | 431.51 ± 185.55 | 1.6 | 175.7 | 0.28 | 0.91 | 0.13 | 0.94 |
| 1214 | 379.77 ± 239.26 | 1.2 | 209.9 | 0.28 | 0.91 | 0.13 | 0.94 |
| 1220 | 438.55 ± 219.28 | 3.8 | 342.3 | 0.18 | 0.92 | 0.11 | 0.92 |
| 1221 | 422.53 ± 152.11 | 2.7 | 342.6 | 0.43 | 1.10 | 0.08 | 1.12 |
| 1222 | 422.40 ± 190.08 | 3.0 | 342.5 | 0.43 | 1.10 | 0.08 | 1.12 |


**3.3 Comparison with IPCC method**
The direct CH$_4$ emissions from WWTP were calculated using the IPCC method, with
data sourced from the Urban Drainage Statistical Yearbook of 2017. By applying the
formula to the basic information of the WWTP outlined in the yearbook, the emission
flux of 213.95 ± 128.37 t a$^{-1}$ was determined, with the uncertainty derived from the data
summarized in the research (Lin et al., 2021). Figure 5 shows the contrast between the
emission inversion results from the monitoring experiment and the emission inventory.
The uncertainty of the inversion results was determined by the uncertainties in wind
speed, wind direction, and instrument measurements. The summer average inversion





emission flux $(603.33 \pm 152.66 \text{ t a}^{-1})$ was calculated to be 2.8 times that of the inventory,
and the winter average $(418.95 \pm 187.59 \text{ t a}^{-1})$ was twice as much. It is posited that the
discrepancy may stem from significant uncertainties in the emission factors associated
with the WWTPs or the lack of updated activity level data, as the statistical yearbook
provided data only up to 2017, the emission inventory might have underestimated the
actual emissions.
Furthermore, other studies have also investigated the comparison between $CH_4$
emissions obtained from different measurement methods at WWTPs and IPCC
inventory estimates. The majority of these studies indicate that the measured
values exceed the inventory values. Wang et al. (2021) conducted a measurement-
based assessment of $CH_4$ emissions $(46.58 \text{ t a}^{-1})$ in Wuhu City, revealing a 46.71 %
higher than those calculated using the IPCC method. Moore et al. (2023)
employed mobile monitoring to evaluate $CH_4$ emissions at 63 WWTPs across the
United States. The study showed that the estimates based on the IPCC guidelines
underestimated the emissions from most of the measured plants. Specifically, $CH_4$
emissions from centrally treated domestic wastewater in the U.S. amount to $4.64 \times 10^5$ t
$a^{-1}$, which is 1.9 times greater than the EPA inventory. Song et al. (2023) investigated
$CH_4$ emissions from sewer systems and water resource recovery facilities. Utilizing a
collected dataset, they employed the Monte Carlo analysis method to determine the $CH_4$
emissions from municipal wastewater treatment in the U.S. at $(4.36 \pm 2.8) \times 10^5 \text{ t a}^{-1}$.
This value was approximately twice the estimates provided by the IPCC. The lower
estimated results provided by the IPCC method can be attributed to the neglect of
certain potential emission sources from the emission inventories, including emissions
from equipment in sludge treatment facilities and leaks from pressure relief valves.

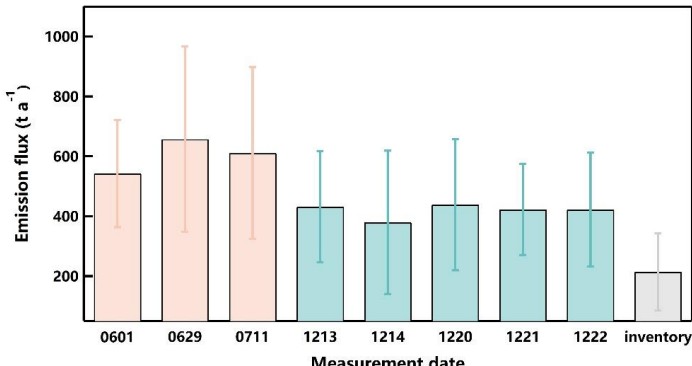

**Figure 5.** Comparison of $CH_4$ emission fluxes from the monitoring experiment and emission inventory in the WWTP.

**3.4 Sensitivity analysis**

In this section, we evaluated the stability of the inversion framework through sensitivity analysis and explored the impact of different point source locations on the inversion of emission concentrations. The precise identification of emission sources can enhance the accuracy of emission flux inversion, making a sensitivity analysis of the source location essential. We applied the method of controlling variables to perform a sensitivity analysis on the location of a single point source. The central position of the plant was taken as the reference origin, the positions of 12 emission sources were determined to analyze the variation in error between measured and simulated concentrations within a 200 m × 200 m range around each emission source. We sequentially modified the source position parameters in the model input to analyze the congruence between the simulated concentrations and the observed measurements, quantifying the fit with RMSE. The change in concentration error serves as an indicator of the accuracy of the emission source localization.

Figures 6 and 7 describe the error variation between monitored and simulated concentrations when the point source location is subject to change within a 200 m × 200 m range from the monitoring experiment on 29th June and 13th December. The error



variation of the remaining days can be seen in Figures S8-S13. The point source

locations simulated based on the inversion framework are mostly in areas with minor

relative concentration errors, which can be considered to have a high reliability in

simulating point source locations. The emission source location errors for the two

experiments are within the ranges of 0.7-1.3 mg m$^{-3}$ and 0.2-0.3 mg m$^{-3}$. The winter

emission source locations exhibit greater stability and accuracy in the inversion results

than the summer ones.

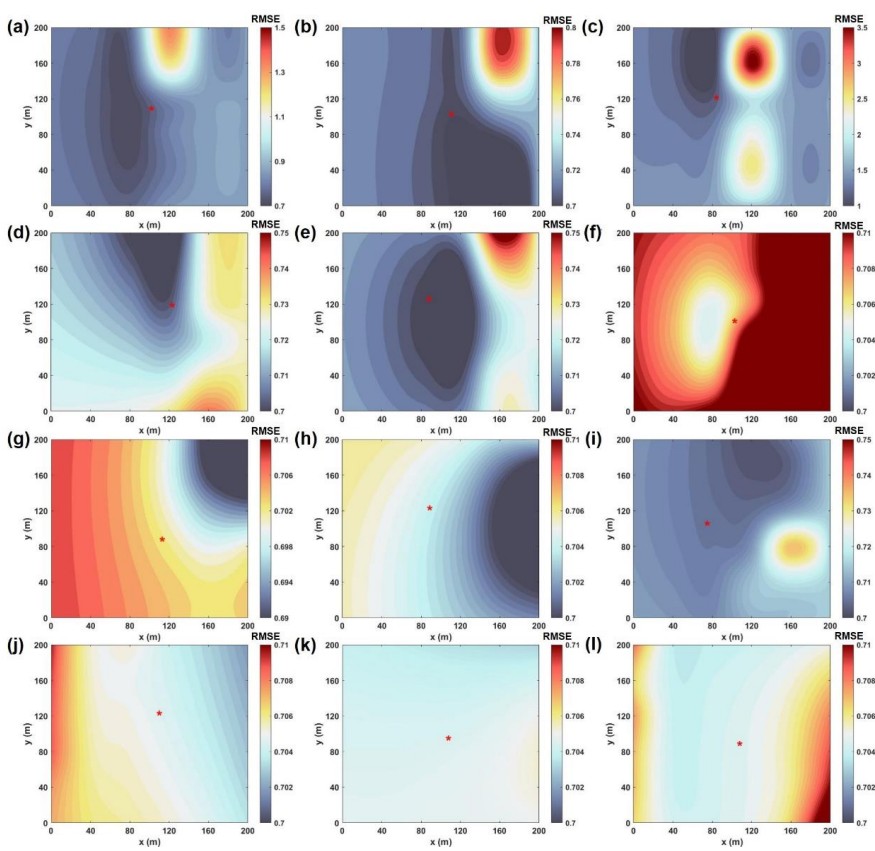

**Figure 6.** RMSE of monitoring simulated concentration changes with the location of

WWTP source on 29$^{th}$ June. The x and y axes denote the horizontal and vertical

distances of the simulated point source from the central point of the WWTP. The

variation in color signifies the alteration in the root mean square error between the





actual monitored and simulated concentrations, with the red star symbolizing the
simulated point source location.

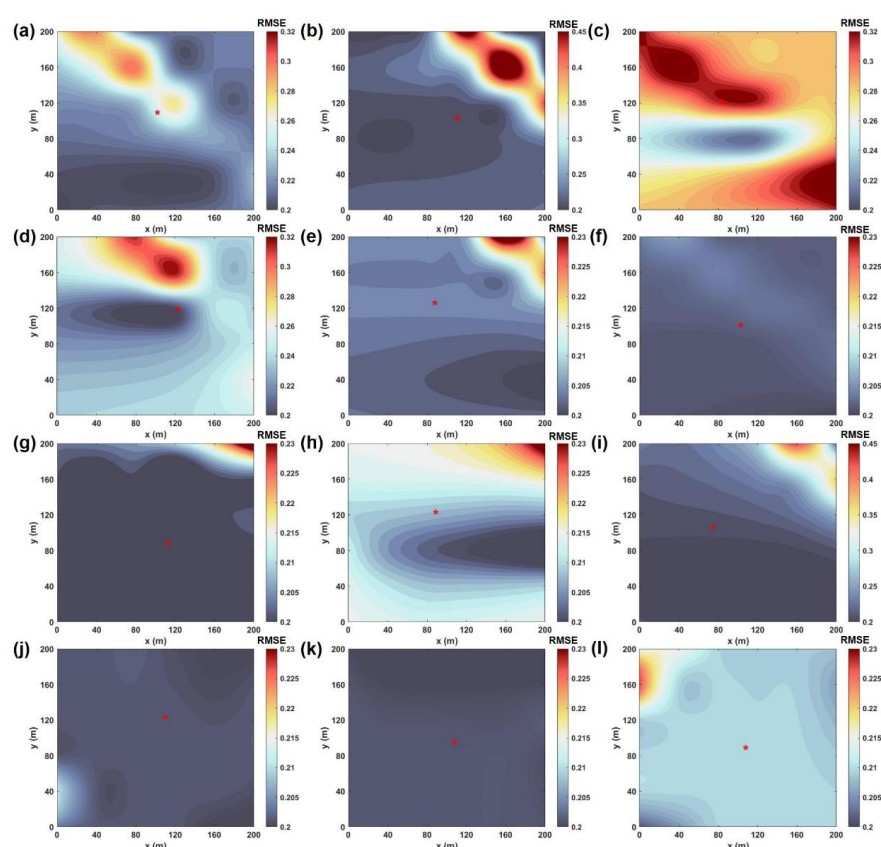

**Figure 7.** RMSE of monitoring simulated concentration changes with the location of
WWTP source on 13<sup>th</sup> December.


**4 Conclusions and outlook**

The study carried out CRDS mobile measurements at a WWTP across the summer
and winter seasons from Hangzhou 2023. By employing a multi-source Gaussian plume
model combined with the genetic algorithm inversion framework, the inversion of $CH_4$
emission fluxes and their source locations was achieved. A sensitivity analysis of the



parameters within the inversion framework was conducted to verify the reliability of the model, offering a strategic approach for the quantification of GHG emissions at the plant scale. The results showed that 12 distinct $CH_4$ emission sources were pinpointed within the facility through the inversion framework. The average $CH_4$ emission flux during the summer was calculated to be $603.33 \pm 152.66$ t a$^{-1}$, and $418.95 \pm 187.59$ t a$^{-1}$ for the winter. The screen and primary clarifier were the main sources, accounting for 55 % of summer and 67 % of winter emissions. When contrasted with bottom-up emission inventory estimates, the summer $CH_4$ inversion emissions were found to be 2.8 times higher, and the winter inversion emissions were twice as much as the inventory values.

The inversion framework is capable of validating emission coefficients in the inventory, identifying emission sources within the plant, and monitoring abnormal emissions. It can be applied to various monitoring systems, such as UAV systems and networks of fixed monitoring stations. We believe that collaborative monitoring by different methods can significantly improve the accuracy of emission fluxes and emission sources inversions. It is suggested that future endeavors focus on refining the inversion framework for broader applicability to various pollutant gases, enhancing the inversion efficiency, and extending the validation of the framework through monitoring experiments in a diverse range of industrial facilities.

*Data availability.* The raw data in this paper can be obtained from the corresponding author upon request.

*Author contributions.* ZW and YZ administrated the project and determined the main goal of this study. ZX, JY and XP designed the methods and planned the campaign. JY, ZX, YY, SZ and BQ performed the measurements. JY wrote the paper with contributions from all co-authors.

*Competing interests.* At least one of the (co-)authors is a member of the editorial board



of Atmospheric Chemistry and Physics.
***Financial support.*** The study has been supported by the National Key Research and
Development Program of China (grant nos. 2022YFC3703500 and 2022YFE0209100),
the National Natural Science Foundation of China (grant no. 42307129), the Key
Research and Development Program of Zhejiang Province (grant nos. 2021C03165 and
2022C03084), the Zhejiang Provincial Natural Science Foundation (grant no.
LZJMZ24D050005), and the Ecological Environment Research and Achievement
Promotion Project of Zhejiang Province (grant nos. 2024XM0053 and 2024XM0052).

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
