# Peer review of "Locating and quantifying CH4 sources within a wastewater"

_EGUsphere, 2024_

## Referee Comment (RC1)

In "Locating and quantifying CH$_4$ sources within a wastewater treatment plant based on mobile measurements" by Junyue Yang, et al., the authors describe an approach to quantify methane emissions from a wastewater treatment plant using a multi-point Gaussian plume model optimized by a genetic inversion algorithm. Atmospheric emissions from WWTPs come from a variety of source types and can be quite variable. This work makes a notable effort to combine previously tested methods to generate plant-level estimates. Overall, this manuscript offers a clear and comprehensive description of the experiment and attempts to position the work in context of the current literature in a compelling manner.

**Comments (general):**

1. Mobile measurements are described as "circling the functional area 1-2 times" for a given experiment. Was one experiment performed per day during the two campaign periods? Is this a sufficient amount of data to generate statistically-based conclusions distinguishing the different sources or seasonal effects? Given the nature and size of the sources, it seems feasible that numerous passes could be necessary to capture any variability in the operations. Some evidence of this is noted in the text (L365) where "the substantial variation in the emissions […] suggests a degree of emission instability" for the aeration tank emission estimates. Based on Figure 5, all of the experiments are within the uncertainty of each other.

2. The characterization of the different sources (i.e., line and point) could be further detailed. Specifically, it would be helpful to see more about how the line source was determined beyond the note in the text ("substantiated by through model validation", L221), perhaps in the Supplement? Also, it seems like the primary clarifiers resemble area sources more than line or point sources. How were they characterized in this work? As multiple point sources?

3. There appears to be no discussion on the possible effect of interferences from surrounding areas. Is there another WWTP neighboring the site? The satellite images show similar equipment or process areas in the adjacent lot outside of the "exterior road". Also, the wind direction in Figures S5, S6, and S7 comes from an area on land with fields and some buildings, also outside of the subject area, which seem worth identifying. If the model or optimization algorithm address these interferences, then it should be explained in text.

4. In Table 1, wind speed on "0711" was listed as 0.9 m s$^{-1}$. Presumably, this is an average value. Was there a trend in the emission estimates relative to wind speed, specifically low-speed winds? Are there datapoints collected when wind speeds were below 0.7 m s$^{-1}$?

5. Discussion on how uncertainty was calculated is brief and non-specific. Further detail on the contributing factors and how the uncertainty was calculated is needed in the text (see L424 for context).

6. Consider emphasizing discussion of tracer flux ratio, also known as "tracer release", as another top-down method that employs mobile measurements. This method was mentioned specifically in text as an example, but it is noted throughout other publications (Delre et al., 2017; Moore et al. 2023; Yacovitch, et al. 2015; von Fischer et al. 2017), including use for validation or comparison with other modelling techniques for similar applications.

7. Using units of t a$^{-1}$ could be useful for comparison-sake (e.g., with the IPCC inventory method), but given the limited number of measurement days, it seems like a significant extrapolation. Other studies present findings in kg h$^{-1}$ (as noted in the introduction) or also

otherwise normalized by other factors. Perhaps consider presenting the results in kg h$^{-1}$ (or similar) with mention of the extrapolation (t a$^{-1}$) to offer the reader a quick comparison to other relevant figures.

**Comments (line-by-line):**

1. L23: replace "emission" with "emissions".
2. L29: consider changing the verb tense of "are" to "is".
3. L57: remove " 's" from "(IEA)'s" or change sentence.
4. L63: clarify the meaning of "actual emission factors activity data" depending on the intent of the sentence. For example, "activity data used for actual emission factors". Or if that's not the intention, then perhaps, "activity-based emission factors".
5. L137: change "Spectroscopy" to "Spectrometer" or add "monitoring system" as done in the Instrumentation section and elsewhere.
6. L147: is this "1-2 times" *per day*? Does each day comprise of one experiment? This should be addressed in the text.
7. L148: is a "set" equivalent to an experiment?
8. L148: what deems these datasets "valid" in this context?
9. L148: what happened to the other two days/experiments?
10. L157: were the less abundant isotopologues of $CH_4$ or $CO_2$ used in the data analysis? If not, it should be stated.
11. L163: how was the probe mounted "on" the roof and how did that position "mitigate the effect of vehicular emissions"? This should be addressed in the text.
12. L166: similar to the previous question, how was the meteorological unit mounted to the roof? Given the additional on-site wind stations, this may not be a critical point, but positioning of the wind unit close to the vehicle body will affect the measurement accuracy.
13. L169: specific the manufacturer of the SWS-500.
14. L242: make mg/s into mg s$^{-1}$ for consistency.
15. L308: what does "the similarity of concentration measurement methods" mean? Wasn't the same method applied to the whole dataset? Or is this a reference to the roads driven? The wind direction is clearly different. This should be addressed in the text.
16. L414: are these average values? Or is there only one estimate determined per day? This should be addressed in the text.
17. L414: while it's relatively clear from context what the labels in this table mean (i.e., Q, Ws, W$_d$, $\gamma$, $\alpha$) some more detail in the caption would be helpful.
18. L424: need more detail on the uncertainty estimates in the text. Is this a confidence interval?
19. L509-511: it is an excellent point that collaborative monitoring offers significant advantages.

---

## Author Comment (AC1)

**Reply to comments on "Locating and quantifying CH₄ sources within a wastewater treatment plant based on mobile measurements" by Yang et al.**

**Reply to Anonymous Referee #2**

In "Locating and quantifying CH$_4$ sources within a wastewater treatment plant based on mobile measurements" by Junyue Yang, et al., the authors describe an approach to quantify methane emissions from a wastewater treatment plant using a multi-point Gaussian plume model optimized by a genetic inversion algorithm. Atmospheric emissions from WWTPs come from a variety of source types and can be quite variable. This work makes a notable effort to combine previously tested methods to generate plant-level estimates. Overall, this manuscript offers a clear and comprehensive description of the experiment and attempts to position the work in context of the current literature in a compelling manner.

**Responses:**

We truly appreciate the constructive comments and suggestions raised by the reviewer. Those comments are valuable and very helpful for improving our paper, as well as the important guiding significance to our studies. Below, we provide a point-by-point response to individual comment. And we also polish the content of the manuscript. The responses are shown in plain font, and the added/rewritten parts are presented in italics.

**Comments (general)**

**1.** Mobile measurements are described as "circling the functional area 1-2 times" for a given experiment. Was one experiment performed per day during the two campaign

periods? Is this a sufficient amount of data to generate statistically-based conclusions distinguishing the different sources or seasonal effects? Given the nature and size of the sources, it seems feasible that numerous passes could be necessary to capture any variability in the operations. Some evidence of this is noted in the text (L365) where "the substantial variation in the emissions […] suggests a degree of emission instability" for the aeration tank emission estimates. Based on Figure 5, all of the experiments are within the uncertainty of each other.

**Responses and Revisions:**

Thank you for the suggestion. The description of "circling the functional area 1-2 times" refers to 1-2 rounds of the entire road measurements along external roads and internal functional areas roads of the WWTP. In the actual measurements, we conducted repeated mobile measurements and fixed measurements on roads with high concentrations to calculate average concentration data, investigating the different sources and seasonal effects of $CH_4$ emissions from the WWTP. Each day of the experiment eventually forms a set of complete data. Based on the conclusion that there is a certain degree of emission instability in the aeration tank emission estimates, we agree that as more days of mobile measurement experiments are added, the monitoring accuracy will be elevated. The following text has been added in the text (Line128-132):

*"We performed one monitoring experiment per day, with each experiment entailing 1-2 rounds of mobile measurements along external roads and internal functional areas roads. Additionally, we conducted repeated mobile measurements and fixed measurements on roads with high concentrations to calculate average concentration data".*

**2.** The characterization of the different sources (i.e., line and point) could be further detailed. Specifically, it would be helpful to see more about how the line source was determined beyond the note in the text ("substantiated by through model validation", L221), perhaps in the Supplement? Also, it seems like the primary clarifiers resemble area sources more than line or point sources. How were they characterized in this work? As multiple point sources?

**Responses and Revisions:**

Thank you for the suggestion. The determination of line sources was previously described in section 2.4.2 "General Finite Line Source Model" and supplemented with Figure S1. By comparing the actual measurement data and model simulation data, we identified multiple point sources and line source diffusion patterns. The comparison is depicted in the revised Figures S1, with Figure S1(a) showing the comparison between $CH_4$ concentration measurements for Aeration Tank ① and point source model simulation on 29th June, and Figure S1(b) comparing $CH_4$ concentration measurements with the line source model simulation on a road between the Screen ① and the Primary Clarifier ①. Figure S1 illustrates that the $CH_4$ concentration curve presents distinct peak distributions in point source diffusion patterns, while the $CH_4$ concentration distribution in line source diffusion patterns consistently maintains higher levels. Considering the $CH_4$ emissions from the primary clarifiers of the WWTP as point and line source diffusion patterns rather than area sources allows for a better integration with mobile measurements to pinpoint the specific locations of $CH_4$ leaks in the WWTP, supporting the development of targeted $CH_4$ reduction strategies. The additional content is as follows (Line193-198):

*"Based on the comparison of measurement and model simulation results (Fig. S1), it is determined that the plant exhibits multiple point sources and line source diffusion patterns. Fig. S1 illustrates that the $CH_4$ concentration curve presents distinct peak distributions in point source diffusion patterns, while the $CH_4$ concentration distribution in line source diffusion patterns consistently maintains higher levels".*

[Figure]

**Figure S1.** Comparison of CH$_4$ measurement and model simulation. (a) CH$_4$ concentration measurements for Aeration Tank ① and point source model simulation on 29$^{th}$ June (b) CH$_4$ concentration measurements and the line source model simulation on a road between the Screen ① and the Primary Clarifier ①.

**3.** There appears to be no discussion on the possible effect of interferences from surrounding areas. Is there another WWTP neighboring the site? The satellite images show similar equipment or process areas in the adjacent lot outside of the "exterior road". Also, the wind direction in Figures S5, S6, and S7 comes from an area on land with fields and some buildings, also outside of the subject area, which seem worth identifying. If the model or optimization algorithm address these interferences, then it should be explained in text.

**Responses and Revisions:**

Thank you for the suggestion regarding external source interference. There are expansion facilities of the WWTP near the monitoring site. However, we were unable to conduct internal roads measurements due to restricted access to the expanded area. We performed mobile measurements along the perimeter of the expansion and found no significant $CH_4$ emissions. Therefore, we concluded that the expansion facilities do not impact the inversion results. When employing the multi-source Gaussian diffusion model combined with a genetic algorithm framework for iterative optimization to pinpoint point sources, we were able to locate external sources. We have considered external sources during the prior inversion process, which were not explicitly mentioned in the previous manuscript. We have recognized the necessity for a detailed explanation of this aspect, and we have added Fig. S2 related to external sources in the supplement. Fig. S2 (a) and (c) depict the emission sources localization on $29^{th}$ June and $13^{th}$ December, revealing that external sources are primarily located along the main roadway to the south of the WWTP. Moreover, the estimated emissions vary across different days, which indicates that external source emissions are influenced by traffic vehicles. Fig. S2 (b) and (d) compare measured and simulated $CH_4$ concentrations before and after accounting for external sources, demonstrating that simulations are significantly closer to the measurements when external sources are included. Therefore, we conclude that considering external sources during the inversion process can enhance the accuracy of the results. The additional content is as follows (Line379-387):

*"In addition, when employing the multi-source Gaussian diffusion model combined with a genetic algorithm framework for iterative optimization to pinpoint point sources, we were able to locate external sources. As shown in Fig. S2 (a) and (c), external sources are primarily located along the main roadway to the south of the WWTP. Moreover, the estimated emissions vary across different days, which indicates that external source emissions are influenced by traffic vehicles. Fig. S2 (b) and (d) compare measured and simulated CH4 concentrations before and after accounting for external sources, demonstrating that simulations are significantly closer to the measurements when external sources are included".*

[Figure]

**Figure S2.** The emission sources localization on 29th June (a) and 13th December (c). And the comparison of CH4 measurement and model simulation (considering external sources) on 29th June (b) and 13th December (d).

**4.** In Table 1, wind speed on "0711" was listed as 0.9 m s$^{-1}$. Presumably, this is an average value. Was there a trend in the emission estimates relative to wind speed, specifically low speed winds? Are there datapoints collected when wind speeds were below 0.7 m s$^{-1}$?

**Responses and Revisions:**

Thank you for raising the issue regarding wind speeds. Wind speed data in Table1 are the average wind speed during the monitoring time. We compared the emission estimates at different wind speeds during the summer and winter seasons. In the summer data, the lowest wind speed recorded was 0.9 m s$^{-1}$ on 11th July, with emission estimates higher than those on 1st June but lower than those on 29th June. In the winter data, the lowest wind speed was 1.2 m s$^{-1}$ on 14th December, and the estimated emissions were lower than those on the other days. Thus, we conclude that there is no significant trend

in emission estimates under low wind speed conditions. Furthermore, we have reviewed and organized the wind speed data and found that datapoints were collected when the wind speeds were below 0.7 m s$^{-1}$. This was due to the fact that the mobile monitoring system was deployed for internal functional area monitoring, where the monitoring vehicle was in close proximity to the functional area facilities. Consequently, concentration data could still be detected even at the conditions of low wind speeds. The additional content is as follows (Line416-419):

*"At lower wind speeds, the CH$_4$ emissions show only slight differences when compared to emissions on days with higher wind speeds. This suggests that the inversion results are less influenced by wind speed and are primarily associated with seasonal variations".*

**5.** Discussion on how uncertainty was calculated is brief and non-specific. Further detail on the contributing factors and how the uncertainty was calculated is needed in the text (see L424 for context).

**Responses and Revisions:**

Thank you for the suggestion. We have incorporated detailed information in the paper to enhance the understanding of the uncertainty analysis. The added text is as follows:

Line288-299: *"To quantify the uncertainty in the inversion results, we have considered the uncertainties associated with the input parameters of the inversion model, including wind speed, wind direction, and instrument measurements. The uncertainty in the CH$_4$ emission fluxes ($\varepsilon_t$) is derived using the error propagation formula as follows:*

$$\varepsilon_t = \sqrt{\varepsilon_s{}^2 + \varepsilon_d{}^2 + \varepsilon_m{}^2} \qquad (7)$$

*$\varepsilon_s$ and $\varepsilon_d$ denote the uncertainties in wind speed and direction, which are determined by the standard deviation of the wind speed and direction measurements from two fixed meteorological stations during the observation period. $\varepsilon_m$ represents the uncertainty in instrumental measurements. This uncertainty is derived from data*

*provided by the manufacturer Picarro, which indicates a concentration measurement uncertainty of approximately 1 ppb for a 10-second integration time (Picarro 2010)".*

*Line468-472: "The uncertainty of the inversion results was determined by accounting for the uncertainties in wind speed, wind direction, and instrument measurements, following the method presented in Section 2.4.5. The uncertainties in emission fluxes inversion range from 33% to 63% on individual days. Notably, the uncertainty associated with wind speed contributes approximately 44% to 94% of the uncertainty range".*

**6.** Consider emphasizing discussion of tracer flux ratio, also known as "tracer release", as another top-down method that employs mobile measurements. This method was mentioned specifically in text as an example, but it is noted throughout other publications (Delre et al., 2017; Moore et al. 2023; Yacovitch, et al. 2015; von Fischer et al. 2017), including use for validation or comparison with other modelling techniques for similar applications.

**Responses and Revisions:**

Thank you for the suggestion. We have added tracer gas dispersion method content to Section 3.2. The added text is as follows (Line437-460):

*"An alternative top-down approach known as the tracer gas dispersion method (TDM), which has been applied to estimate $CH_4$ emissions from city streets (von Fischer et al., 2017; Weller et al., 2018), WWTPs (Yoshida et al., 2014; Delre et al., 2017; Delre et al., 2018), biogas plants (Reinelt et al., 2017; Scheutz et al., 2019; Fredenslund et al., 2023) and landfills (Rees-White et al., 2019; Kissas et al., 2022). TDM involves releasing tracer gases like nitrous oxide and acetylene near the source and measuring their concentrations along with $CH_4$ downwind using a mobile platform. The similar diffusion patterns of $CH_4$ and the tracer gases result in a stable concentration ratio after atmospheric mixing, enabling the calculation of $CH_4$ emission rates with a better accuracy (Mønster et al., 2014).*

*Moreover, the TDM was employed to validate and compare other model inversion methods. Moore et al. (2023) proposed that the Gaussian dispersion modeling demands*

*less experimental equipment, site access, and manpower than the TMD, which allows for swifter data gathering. Yacovitch et al. (2015) utilized a five-day dataset of tracer gas release in the Barnett shale region to evaluate the Gaussian dispersion flux quantification method. The results indicated a 95% confidence interval, with a lower bound factor of 0.334 and an upper bound factor of 3.34. von Fischer et al. (2017) performed three controlled release experiments to validate the reliability of leak rate algorithm in Fort Collins, CO, which indicated a very significant correlation between known and estimated leak rates ($p < 0.0001$, $r^2 = 0.43$). Compared to the method employed in the study, the TDM offers advantages such as simpler formula calculations. Nonetheless, it also presents several drawbacks, including complex experimental procedures, safety hazards associated with the release of tracer gases, which require access permits from industrial facilities".*

**7.** Using units of t a$^{-1}$ could be useful for comparison-sake (e.g., with the IPCC inventory method), but given the limited number of measurement days, it seems like a significant extrapolation. Other studies present findings in kg h$^{-1}$ (as noted in the introduction) or also otherwise normalized by other factors. Perhaps consider presenting the results in kg h$^{-1}$ (or similar) with mention of the extrapolation (t a$^{-1}$) to offer the reader a quick comparison to other relevant figures.

**Responses and Revisions:**

Thank you for the suggestion on the emission flux unit. We have revised the unit of emission flux of the WWTP with kg h$^{-1}$, including a mention of the extrapolation to (t a$^{-1}$). The units of t a$^{-1}$ have been maintained for the figure and text in section 3.3 to facilitate a clearer contrast with the emission inventory.

**Comments (line-by-line)**

**1. L23:** replace "emission" with "emissions".

**Responses and Revisions:**

Thank you for the suggestion. We have revised it to *"emissions"*.

**2. L29:** consider changing the verb tense of "are" to "is".

**Responses and Revisions:**

Thank you for the suggestion. We have changed the verb tense to *"is"*.

**3. L57:** remove " 's" from "(IEA) 's" or change sentence.

**Responses and Revisions:**

Thank you for the suggestion. We have revised it to *"(IEA)"*.

**4. L63:** clarify the meaning of "actual emission factors activity data" depending on the intent of the sentence. For example, "activity data used for actual emission factors". Or if that's not the intention, then perhaps, "activity-based emission factors".

**Responses and Revisions:**

Thank you for the suggestion. We have acknowledged that our previous phrasing was ambiguous and could lead to misinterpretation. We intended to convey that the activity data are based on actual emission factors. We have revised it to *"activity data used for actual emission factors"*.

**5. L137:** change "Spectroscopy" to "Spectrometer" or add "monitoring system" as done in the Instrumentation section and elsewhere.

**Responses and Revisions:**

Thank you for the suggestion. We have changed it to *"Spectrometer"*.

**6. L147:** is this "1-2 times" per day? Does each day comprise of one experiment? This should be addressed in the text.

**Responses and Revisions:**

Thank you for the suggestion. we have recognized the necessity for a more detailed experimental description. We performed one monitoring experiment per day, with each experiment entailing 1-2 rounds of monitoring along external roads and internal

functional areas roads. During each experiment, we conducted repeated mobile measurements and fixed measurements on roads with high concentrations to enhance the accuracy of concentration monitoring. The following text has been added in the text (Line128-132):

*"We performed one monitoring experiment per day, with each experiment entailing 1-2 rounds of mobile measurements along external roads and internal functional areas roads. Additionally, we conducted repeated mobile measurements and fixed measurements on roads with high concentrations to calculate average concentration data".*

**7. L148:** is a "set" equivalent to an experiment?

**Responses and Revisions:**

The term "a set of data" is composed of external road concentration data and internal functional area concentration data. During each experiment, we conducted repeated mobile measurements and fixed measurements on roads with high concentrations to enhance the accuracy of concentration monitoring. Each day of the experiment eventually forms a set of valid data.

We have recognized that this expression can be confusing. Thus, we have changed this in the revised manuscript (Line132-134): *"Over 10 days of experiments from June to December 2023, we obtain 8 days of complete monitoring data, including 3 days in summer and 5 days in winter".*

**8. L148:** what deems these datasets "valid" in this context?

**Responses and Revisions:**

Thank you for the suggestion. Valid data refers to the complete $CH_4$ concentration data from the WWTP, including both external road data and internal functional areas data of the WWTP.

We recognize that this expression can be confusing. Thus, we have changed this in the revised manuscript (Line132-134): *"Over 10 days of experiments from June to*

*December 2023, we obtain 8 days of complete monitoring data, including 3 days in summer and 5 days in winter".*

**9. L148:** what happened to the other two days/experiments?

**Responses and Revisions:**

Thank you for the suggestion. The other two days of experiments were conducted on 28th June and 12th December However, due to the internal maintenance of the wastewater treatment facilities, which rendered certain roads inaccessible, we could not obtain comprehensive concentration data from the plant, and the quantification of emission sources was not carried out. The additional content is as follows (Line134-135):

*"On the other two experimental days, internal facility maintenance restricted access to certain roads, resulting in incomplete monitoring data".*

**10. L157:** were the less abundant isotopologues of $CH_4$ or $CO_2$ used in the data analysis? If not, it should be stated.

**Responses and Revisions:**

Thank you for the suggestion. $CH_4$ and $CO_2$ isotopologues data were not used in the data analysis. *The sentence "The CRDS analyzer can be used to measure $^{12}CO_2$, $^{13}CO_2$, $^{12}CH_4$, $^{13}CH_4$ and $H_2O$"* has been deleted. We have changed this in the revised manuscript (Line150-151):

*"In this study, the CRDS analyzer was placed inside the monitoring vehicle to measure $CH_4$ concentrations in the WWTP".*

**11. L163:** how was the probe mounted "on" the roof and how did that position "mitigate the effect of vehicular emissions"? This should be addressed in the text.

**Responses and Revisions:**

Thank you for the suggestion on the instrument system description. We have realized that the installation position of the sampling probe was not accurately described. We have changed this in the revised manuscript (Line151-152):

*"The sampling probe was placed near the roof along the window of the car to avoid interference from vehicle exhaust due to the low position".*

**12. L166:** similar to the previous question, how was the meteorological unit mounted to the roof? Given the additional on-site wind stations, this may not be a critical point, but positioning of the wind unit close to the vehicle body will affect the measurement accuracy.

**Responses and Revisions:**

Thank you for the suggestion on the description of the instrument system. The meteorological instrument was adsorbed on the roof of the vehicle through the magnet at the bottom. We agree that positioning of the wind unit close to the vehicle body will affect the measurement accuracy. Consequently, the meteorological instrument on the vehicle is mainly regarded as a reference for the instantaneous wind direction during mobile measurements. The meteorological data used as input for the model simulations are obtained from portable meteorological stations.

**13. L169:** specific the manufacturer of the SWS-500.

**Responses and Revisions:**

Thank you for the suggestion. We have added specific content for SWS-500 (Hangzhou Pengpu Technology).

**14. L242:** make mg/s into mg s$^{-1}$ for consistency.

**Responses and Revisions:**

Thank you for the suggestion. We have revised it to "mg s$^{-1}$".

**15. L308:** what does "the similarity of concentration measurement methods" mean? Wasn't the same method applied to the whole dataset? Or is this a reference to the roads driven? The wind direction is clearly different. This should be addressed in the text.

**Responses and Revisions:**

Thank you for pointing out the error in the sentence. The method for measuring concentration is the same. We have changed this in the revised manuscript (Line311):

*"Due to the consistency of concentration measurement methods, we chose 29th June and 13th December as a typical example"*.

**16. L414:** are these average values? Or is there only one estimate determined per day? This should be addressed in the text.

**Responses and Revisions:**

Thank you for the suggestion. Wind speed and direction data are the average wind speed and direction during the monitoring time, and the $CH_4$ emission fluxes are obtained by the inversion of the average concentration, wind speed and direction data as input of the model, which are one estimate determined per day. The following text has been revised and added in the text (Line398-403):

*"Table 1 displays the $CH_4$ emission fluxes (Q), Wind speed and direction data ($W_s$, $W_d$), the horizontal diffusion coefficient ($\gamma_1$, $\alpha_1$) and the vertical diffusion coefficient ($\gamma_2$, $\alpha_2$) from the 8-day monitoring experiment. Wind speed and direction data are the average wind speed and direction during the monitoring time, and the $CH_4$ emission fluxes are obtained by the inversion of the average concentration, wind speed and direction data as input of the inversion framework"*.

**17. L414:** while it's relatively clear from context what the labels in this table mean (i.e., Q, Ws, Wd, $\gamma$, $\alpha$) some more detail in the caption would be helpful.

**Responses and Revisions:**

Thank you for the suggestion to improve the caption for Table 1. We have revised the title (Line433-435) to *"$CH_4$ emission fluxes (Q), Wind speed and direction data ($W_s$, $W_d$), the horizontal diffusion coefficient ($\gamma_1$, $\alpha_1$) and the vertical diffusion coefficient ($\gamma_2$, $\alpha_2$) from the 8-day monitoring experiment"*.

**18. L424:** need more detail on the uncertainty estimates in the text. Is this a confidence interval?

**Responses and Revisions:**

Thank you for the suggestion regarding the section on uncertainty analysis. We have incorporated additional details to enhance the understanding of the uncertainty analysis. The determination of the numerical values is not based on confidence intervals, but rather calculations derived from wind speed, wind direction, and instrumental uncertainties. The added text is as follows:

Line288-299: *"To quantify the uncertainty in the inversion results, we have considered the uncertainties associated with the input parameters of the inversion model, including wind speed, wind direction, and instrument measurements. The uncertainty in the CH$_4$ emission fluxes ($\varepsilon_t$) is derived using the error propagation formula as follows:*

$$\varepsilon_t = \sqrt{\varepsilon_s{}^2 + \varepsilon_d{}^2 + \varepsilon_m{}^2} \qquad (7)$$

*$\varepsilon_s$ and $\varepsilon_d$ denote the uncertainties in wind speed and direction, which are determined by the standard deviation of the wind speed and direction measurements from two fixed meteorological stations during the observation period. $\varepsilon_m$ represents the uncertainty in instrumental measurements. This uncertainty is derived from data provided by the manufacturer Picarro, which indicates a concentration measurement uncertainty of approximately 1 ppb for a 10-second integration time (Picarro 2010)"*.

Line468-472: *"The uncertainty of the inversion results was determined by accounting for the uncertainties in wind speed, wind direction, and instrument measurements, following the method presented in Section 2.4.5. The uncertainties in emission fluxes inversion range from 33% to 63% on individual days. Notably, the uncertainty associated with wind speed contributes approximately 44% to 94% of the uncertainty range"*.

**19. L509-511:** it is an excellent point that collaborative monitoring offers significant

**Responses and Revisions:**

Thank you for the suggestion. We have changed this in the revised manuscript (Line550-552):

*"We believe that it is an excellent point that collaborative monitoring offers significant improvements in the accuracy of emission fluxes and source inversion"*.

**References**

Delre, A., Mønster, J., and Scheutz, C.: Greenhouse gas emission quantification from wastewater treatment plants, using a tracer gas dispersion method, Sci. Total. Environ., 605-606, 258-268, http://dx.doi.org/10.1016/j.scitotenv.2017.06.177, 2017.

Delre, A., Mønster, J., Samuelsson, J., Fredenslund, A. M., and Scheutz, C.: Emission quantification using the tracer gas dispersion method: the influence of instrument, tracer gas species and source simulation, Sci. Total. Environ., 634, 59-66, https://doi.org/10.1016/j.scitotenv.2018.03.289, 2018.

Fredenslund, A. M., Gudmundsson, E., Falk, J. M., and Scheutz, C.: The Danish national effort to minimise methane emissions from biogas plants, Waste Manag., 157, 321-329, https://doi.org/10.1016/j.wasman.2022.12.035, 2023.

Kissas, K., Ibrom, A., Kjeldsen, P., and Scheutz, C.: Methane emission dynamics from a Danish landfill: The effect of changes in barometric pressure, Waste Manag., 138, 234-242, https://doi.org/10.1016/j.wasman.2021.11.043, 2022.

Mønster, J., Samuelsson, J., Kjeldsen, P., Rella, C.W., and Scheutz, C.: Quantifying methane emission from fugitive sources by combining tracer release and downwind measurements- A sensitivity analysis based on multiple field surveys, Waste Manag., 34:1416-1428, https://doi.org/10.1016/j.wasman.2014.03.025, 2014.

Moore, D. P., Li, N. P., Wendt, L. P., Castañeda, S. R., Falinski, M. M., Zhu, J.-J., Song, C., Ren, Z. J., and Zondlo, M. A.: Underestimation of sector-wide methane emissions from United States wastewater treatment, Environ. Sci. Technol., 57, 4082-4090, https://doi.org/10.1021/acs.est.2c05373, 2023.

Rees-White, T. C., Mønster, J., Beaven, R. P., and Scheutz, C.: Measuring methane emissions from a UK landfill using the tracer dispersion method and the influence of operational and environmental factors, Waste Manage., 87, 870-882, https://doi.

org/10.1016/j.wasman.2018.03.023, 2019.

Reinelt, T., Delre, A., Westerkamp, T., Holmgren, M.A., Liebetrau, J., and Scheutz, C.: Comparative use of different emission measurement approaches to determine methane emissions from a biogas plant. Waste Manag., 68, 173-185, http://dx.doi.org/10.1016/j.wasman.2017.05.053, 2017.

Scheutz, C., and Kjeldsen, P.: Guidelines for landfill gas emission monitoring using the tracer gas dispersion method. Waste Manage., 85, 351-360, https://doi.org/10.1016/j.wasman.2018.12.048, 2019.

von Fischer, J. C., Cooley, D., Chamberlain, S., Gaylord, A., Griebenow, C. J., Hamburg, S. P., Salo, J., Schumacher, R., Theobald, D., and Ham, J.: Rapid, vehicle-based identification of location and magnitude of urban natural gas pipeline leaks, Environ. Sci. Technol., 51, 4091-4099, https://doi.org/10.1021/acs.est.6b06095, 2017.

Weller, Z. D., Roscioli, J. R., Daube, W. C., Lamb, B. K., Ferrara, T. W., Brewer, P. E., and von Fischer, J. C.: Vehicle-based methane surveys for finding natural gas leaks and estimating their size: Validation and uncertainty, Environ. Sci. Technol., 52, 11922-11930, https://doi.org/10.1021/acs.est.8b03135, 2018.

Yacovitch, T. I., Herndon, S. C., Petron, G., Kofler, J., Lyon, D., Zahniser, M. S., and Kolb, C. E.: Mobile laboratory observations of methane emissions in the Barnett Shale Region, Environ. Sci. Technol., 49, 7889-7895, https://doi.org/10.1021/es506352j, 2015.

Yoshida, H., Mønster, J., and Scheutz, C.: Plant-integrated measurement of greenhouse gas emissions from a municipal wastewater treatment plant, Water Res., 61, 108-118, http://dx.doi.org/10.1016/j.watres.2014.05.014, 2014.

---

## Author Comment (AC2)

**Reply to comments on "Locating and quantifying CH₄ sources within a wastewater treatment plant based on mobile measurements" by Yang et al.**

**Reply to Anonymous Referee #1**

This study conducted $CH_4$ mobile measurements in a wastewater treatment plant in summer and winter of 2023 and utilized a multi-source Gaussian plume model along with a genetic algorithm inversion framework to locate major sources within the plant and quantify the corresponding $CH_4$ emission fluxes. Similar to previous studies, they found that emission estimates based on their inversion framework were higher than those estimated using traditional IPCC methods. They also found that the emissions are higher during summer than winter. Given the important role of $CH_4$ in climate change, these types of studies are essential. So, the paper is within the scope of ACP. However, the paper lacks some key details which makes it hard to understand the experimental design and the inversion framework.

**Responses:**

We thank the reviewer for the constructive suggestions and comments concerning our manuscript. These comments are invaluable and greatly assist in enhancing our paper. Below, we present a point-by-point response to each individual comment. And we also polish the content of the manuscript. The responses are shown in plain font, and the added/rewritten parts are presented in italics.

**Major comments**

**1. Figure 1:** Can you please explain how the numbers for different components are defined in Figure 1? I see secondary clarifiers 1, 2, and 5 but not 3 and 4. It would also be helpful to show the roads on which you drove the mobile van.

**Responses and Revisions:**

  Thank you for the suggestion. We initially assigned numbers to the facilities in sequential order based on their quantity. The designation "Secondary Clarifier ⑤" was chosen because it is adjacent to "Primary Clarifier ⑤" and "Aeration Tank ⑤". We have acknowledged that this naming convention may cause misunderstandings. Consequently, we have revised the label in Figure 1 from "Secondary Clarifier ⑤" to "Secondary Clarifier ③". Furthermore, we have modified Figure 1 to show the roads on which we drove the mobile van.

**2. Lines 147-149:** Why were two days of data left out? How were the monitoring days determined? What were the meteorological conditions during the measurement days and were those conditions representative of typical summer and winter conditions?

**Responses and Revisions:**

  The other two days of experiments were conducted on 28th June and 12th December. However, due to the internal maintenance of the wastewater treatment facilities, which rendered certain roads inaccessible, we could not obtain comprehensive concentration data from the plant, and the quantification of emission sources was not carried out. About monitoring days,the monitoring experiment was carried out when the WWTP was conveniently opened in summer and winter and there was no significant precipitation throughout the day. Compared with the historical weather data, the meteorological conditions of the monitoring date are in line with the typical weather conditions in summer and winter. The additional content is as follows (Line132-135):

*"Over 10 days of experiments from June to December 2023, we obtain 8 days of complete monitoring data, including 3 days in summer and 5 days in winter. On the other two experimental days, internal facility maintenance restricted access to certain roads, resulting in incomplete monitoring data".*

**3. Lines 210-211:** Can you please provide more details about how the initial emission estimates are derived? Are they derived in Section 2.3? Since the sources are so close to each other, there is a high possibility of plumes overlapping with each other and the observed concentrations being affected by multiple sources. Can you please explain how this overlapping issue was addressed for the 12 sources considered in the multi-source Gaussian plume?

**Responses and Revisions:**

Thank you for the suggestion. The initial emission estimates are not derived in Section 2.3. The initial emission estimates are mainly based on the concentration distribution and internal functional area features of the WWTP to define the initial source positions and to approximate the initial emissions using established empirical formula (Weller et al., 2019). We agree that the superposition of 12 sources may introduce discrepancies in the initial predictions. Therefore, these initial source locations and emissions serve as a reference point, with further optimization of the emission source locations and emissions through model inversion. In the implementation of a multi-source Gaussian model inversion, we have taken into consideration the overlapping plumes from point sources, ensuring that concentration levels are accurately represented through spatial superposition. The additional content is as follows (Line211-218):

*"We used the improved empirical equation to estimate the initial emissions of emission sources (von Fisher et al., 2017; Weller et al.,2019). This method was primarily utilized for urban $CH_4$ leakage source emissions estimation (Defratyka et al., 2021; Maazallahi et al., 2020). The empirical equation is as follows:*

$$ln\left(M_{CH_4}\right) = -0.988 + 0.817 \times ln\ (CH_4\ emission\ rate) \qquad (2)$$

*The $M_{CH_4}$ is the maximum enhancement value of $CH_4$ concentration, ppm. The $CH_4$ emission rate represents the $CH_4$ emission flux, L min$^{-1}$".*

**4. Line 223:** What about the emissions upwind (e.g., secondary clarifier 2, power sanitation 1 etc.) of primary qualifier 1? How are those removed from this line source?

**Responses and Revisions:**

In the actual measurements, different points on the road of the WWTP were selected for fixed measurements. By analyzing wind direction and corresponding concentration distribution, we ultimately determined the location of the line source. For the emissions upwind (e.g., secondary clarifier ②, power sanitation ① etc.) of primary qualifier ①, no significant line source leakage distribution was detected when it was located downwind. Thus, we infer that this location is not part of the line source.

**5. Equation (2) and lines 240-253:** How are the values of different parameters determined using the observations?

**Responses and Revisions:**

Thank you for the suggestion. The $CH_4$ concentrations were obtained through mobile measurements using vehicle-mounted CRDS monitoring system. The portable meteorological stations collected data on wind speed and direction, while GPS tracked the mobile paths to pinpoint emission source locations. The diffusion parameter power function expression coefficients were selected based on the atmospheric stability conditions of the day, and finally combine them with Equation (2) to obtain the horizontal and vertical dispersion parameters. The additional content is as follows (Line243-246):

*"During the observation, the $CH_4$ concentrations were obtained through the vehicle-mounted CRDS monitoring system. The portable meteorological stations collected data on wind speed and direction, while GPS tracked the mobile paths to pinpoint emission source locations".*

**Minor comments**

**1. Line 26:** What do you mean by "emission data" here? Activity data, emissions factors?

**Responses and Revisions:**

Thank you for the correction of the "emission data" description. The meaning of "emission data" here is activity data We have revised it to *"activity data"*.

**2. Line 28:** Suggest replacing "in combination with" by "using".

**Responses and Revisions:**

Thank you for the suggestion. We have revised it to *"using"*.

**3. Line 35:** Since measurements are done only during 10 days, I recommend reporting the emissions in tons/day rather than tons/annum.

**Responses and Revisions:**

Thank you for the suggestion on the emission flux units. We have revised the units of emission flux of the WWTP with kg h$^{-1}$, as this unit is more suitable for short-term measurements. However, the units of t a$^{-1}$ have been maintained for the figure and text in section 3.3 to facilitate a clearer contrast with the emission inventory.

**4. Line 63:** Suggest adding "," after "emission factors" because emission factors and activity data are different parameters.

**Responses and Revisions:**

Thank you for the suggestion. We have acknowledged that our previous phrasing was ambiguous and could lead to misinterpretation. We intended to convey that the activity data are based on actual emission factors. We have revised it to *"activity data used for actual emission factors"*.

**5. Line 195-196:** Please mention the TOW value deduced from the workbook.

**Responses and Revisions:**

Thank you for the suggestion. The content of TOW value is added as follows (Line176-178):

"*The Total Organic Waste (TOW) is calculated by the amount of treated water and COD influent concentration of the WWTP provided in the yearbook*".

**6. Lines 220-221:** Screen 1 and primary qualifier 1 are located diagonally from each other. Can you mark this road in Figure 1?

**Responses and Revisions:**

Thank you for the suggestion on Figure 1. We have modified Figure 1 to mark the line source.

[Figure]

Figure 1. Distribution of functional areas of the WWTP. The yellow mark represents the simulated location of the line source. Solid lines show the roads measured by the mobile vehicle. Map data are from ESRI.

**7. Lines 316-317:** Were higher background concentrations in December due to shallower boundary layer?

**Responses and Revisions:**

Thank you for the suggestion regarding background concentrations. We agree that the higher background concentrations observed in December are primarily due to the

lower atmospheric boundary layer. The variation in atmospheric boundary layer height significantly affects the $CH_4$ concentration. During summer, due to solar radiation and ground heating, the boundary layer height tends to be relatively high. This enhances the dispersion capacity of $CH_4$ in the atmosphere, leading to a corresponding decrease in the concentrations. In contrast, during winter, the boundary layer height is typically lower, which facilitates the retention of $CH_4$ near the surface, resulting in higher ground-level concentrations. The additional content is as follows (Line318-320):

*"The analysis indicates that the shallower boundary layer in winter causes $CH_4$ to accumulate near the surface, resulting in a higher background concentration".*

**8. Lines 361-362 and Figures 3-4:** How are the emission source locations determined? Are they known a priori?

**Responses and Revisions:**

Thank you for the inquiry regarding the location of the emission sources. The positions of the emission sources were not known a priori. We initially identified the source locations through the concentration distributions obtained from mobile measurements. Subsequently, we utilized model simulations to further refine and optimize the source locations.

**References**

Defratyka, S. M., Paris, J. D., Yver-Kwok, C., Fernandez, J. M., Korben, P., and Bousquet, P.: Mapping urban methane sources in Paris, France, Environ. Sci. Technol., 55, 8583-8591, https://doi.org/10.1021/acs.est.1c00859, 2021.

Maazallahi, H., Fernandez, J. M., Menoud, M., Zavala-Araiza, D., Weller, Z. D., Schwietzke, S., von Fischer, J. C., Denier van der Gon, H., and Röckmann, T.: Methane mapping, emission quantification, and attribution in two European cities: Utrecht (NL) and Hamburg (DE), Atmos. Chem. Phys., 20, 14717-14740, https://doi.org/10.5194/acp-20-14717-2020, 2020.

von Fischer, J. C., Cooley, D., Chamberlain, S., Gaylord, A., Griebenow, C. J., Hamburg, S. P., Salo, J., Schumacher, R., Theobald, D., and Ham, J.: Rapid, vehicle-based identification of location and magnitude of urban natural gas pipeline leaks, Environ. Sci. Technol., 51, 4091-4099, https://doi.org/10.1021/acs.est.6b06095, 2017.

Weller, Z. D., Yang, D.K., von Fischer, J.C.: An open source algorithm to detect natural gas leaks from mobile methane survey data, PLoS ONE, 14, e0212287. https://doi.org/10.1371/journal.pone.0212287, 2019.